# Improvement of the Rubbing Fastness of Cotton Fiber in Indigo/Silicon Non-Aqueous Dyeing Systems

**DOI:** 10.3390/polym11111854

**Published:** 2019-11-11

**Authors:** Yuni Luo, Liujun Pei, Hongjuan Zhang, Qi Zhong, Jiping Wang

**Affiliations:** 1Engineering Research Center for Eco-Dyeing and Finishing of Textiles, Zhejiang Sci-Tech University, Hangzhou 310018, China; yuniluo@163.com (Y.L.); qi.zhong@zstu.edu.cn (Q.Z.); 2School of Fashion Engineering, Shanghai University of Engineering Science, Shanghai 201620, China; zhjdhdx@163.com

**Keywords:** cotton fiber, indigo, silicone non-aqueous medium, rubbing fastness

## Abstract

In order to solve the poor rubbing fastness of dyed cotton fiber in the indigo/silicon non-aqueous dyeing system, the process parameters of the silicon non-aqueous dyeing system were optimized. Dyed cotton fiber was post-treated to achieve the optimum dyeing conditions for obtaining a better rubbing fastness. Meanwhile, the dyeing performance of cotton fiber in a traditional water bath and silicon non-aqueous dyeing system was compared. The results showed that the rubbing fastness of dyed cotton fiber in the silicon non-aqueous dyeing system (one dyeing) was lower than that of traditional water bath (twelve cycles), although the color depth of dyed cotton fiber was deeper. For obtaining a good rubbing fastness, the optimum temperature was about 70 °C and the optimal dyeing cycle was one. Moreover, fixing agents can significantly improve the rubbing fastness of dyed cotton fiber. Especially, cationic waterborne polyurethane had an optimal fixing effect on the dyed cotton fiber. Soft finishing would weaken the effect of fixing finishing on the dyed cotton fiber, but the softener can significantly improve the handle of dyed cotton fiber.

## 1. Introduction

Most traditional cowboy wear is made of warp yarns which are dyed with indigo on the sizing and dyeing machine, and then woven into garments [1,2]. Due to the monotonous color of indigo dyed warp yarns, all kinds of denim garments must be treated with some special chemical treatments [3,4,5] to obtain a variety of styles which are required by consumers. This process is complicated and produces serious environmental pollution. If indigo dyeing can be carried out on the loose cotton fibers, a series of cotton yarn with different performance, shape and indigo color would be produced through spinning with the aid of changes in fiber composition, proportion and the spinning method, and then in weaving (tatting or knitting), abundant fabric surface effects could be produced through the selection of yarn and texture variation, and more personalized high-quality new denim products might be developed [6,7,8]. With such an indigo dyed loose cotton, not only the imagination of the cowboy wear designer could be greatly expanded, but also the flexibility of production could be improved. For example, color level change and fade effect of denim would be produced directly by spinning, weaving and a slightly wet process with no or little use of chemicals [9,10]. It is expected to omit or reduce the subsequent chemical wet process and reduce water consumption and pollution emissions for cowboy wear production.

However, it is difficult to obtain the ideal dyeing depth after one dyeing in a water bath due to the very low affinity of indigo dye to cotton fiber [11], and it can only meet the requirements after repeated consecutive dyeing like the traditional dyeing of warp yarns on sizing and dyeing combined machines. Unfortunately, repeated dyeing will result in serious deterioration of the spinnability of cotton fibers, which makes it impossible difficult to spin qualified yarn. Therefore, it is difficult to produce indigo dyed cotton by traditional water bath dyeing methods. In order to solve this problem, we used an ecological non-aqueous medium dyeing technology [12,13,14] and successfully invented a new indigo dyeing technology for loose cotton. In this method, since the indigo leuco is insoluble in non-aqueous medium but has a high affinity to cotton fiber and the non-aqueous medium has good isolation effect and can prevent the oxidation of the leuco [15], a dark indigo dyeing of cotton fiber with 90% dyeing rate can be obtained by one dyeing and the good spinnability of the fiber can be maintained.

In an indigo/silicon non-aqueous dyeing system, cotton fiber can achieve a deep color depth after one dyeing [16], but the rubbing fastness of indigo may be influenced. In this paper, indigo dyeing performance was studied in a silicon non-aqueous dyeing system, which was compared with that in a traditional water bath. Moreover, the rubbing fastness of dyed cotton fiber was improved with some fixing agents. We believe that this dyeing technology makes the indigo dyeing of loose cotton a reality which has been expected for many years in the cowboy product field.

## 2. Experimental

### 2.1. Materials

The combed velveteen cotton fiber (linear density: 1.8 d (denier), length: 28.5 mm, Micronaire: A) was collected from Shaoxing Furun Dyeing and Finishing Co., Ltd. (Shaoxing, China). The molecular structure of granular Indigo (industrial grade) was shown in Figure 1, which was purchased from Changzhou Lantu Chemical Co., Ltd. (Changzhou, China). Siloxane non-aqueous media (decamethyl cyclopentasiloxane, purity > 98%) was purchased from GE Toshiba Silicone Ltd. Sodium dithionite, sodium hydroxide and acetic acid are analytical grade, were purchased from Hangzhou Gaojing Chemical Reagent Co., Ltd. (Hangzhou, China). Fixing agent A (acrylate polymer), B (cationic waterborne polyurethane) and C (non-ionic waterborne polyurethane) were industrial grade and were purchased from Guangzhou Xuqi Chemical Co., Ltd. (Guangzhou, China), Qingyuan Junyu Chemical Co., Ltd. (Qingyuan, China), Dongguan Tiansheng Chemical Co., Ltd. (Dongguan, China), respectively. Softener was purchased Zhejiang Kefeng Silicone Co., Ltd. (Haining, China).

### 2.2. Indigo Dyeing of Cotton Fiber

Pre-treatment of cotton fiber: 1 g of cotton fiber and 0.25 g of sodium hydroxide were added into 50 mL water. The temperature was kept at 90 °C for 60 min. Then cotton fiber was dried after washing at a liquor ratio of 1:20.

Indigo/silicone non-aqueous dyeing system: 1 g of cotton fiber was dyed with 1 g of high concentration of indigo-reduced solution (6 g indigo, 19 g sodium dithionite, 8 g sodium hydroxide and 100 g water were stirring at 60 °C for 20 min), 0.8 g of water in 20 g of silicon non-aqueous media. The dyeing temperature was raised from room temperature to 70 °C at a rate of 2 °C per minute and held for 60 min. 

Whether a multiple dyeing can be achieved in silicon non-aqueous dyeing system, multiple indigo dyeing was investigated. The dyeing method was the same as the silicon non-aqueous dyeing system. Firstly, indigo-reduced solution was divided into two or three parts. Then, 1/2 or 1/3 of indigo-reduced solution was put into dyeing system for 60 min. After oxidization, the remaining 1/2 or 1/3 of the reduced solution was added into dyeing system for another 60 min. 

Traditional water bath dyeing: 1 g of cotton fiber was dyed with 1 g of high concentration indigo-reduced solution and 19 g of water. The dyeing temperature was raised from room temperature to 40 °C, and the cotton fiber was immersed for 1 min, then was padded at the liquid rate of 150% (owf, on weight the fabric) and oxidized in the air for 2 min. 

After dyeing, cotton fiber was oxidized in the air for 12 h, then pickling at room temperature for 5 min with 0.05 g acetic acid and 30 mL water.

### 2.3. Fixing Method

A total of 1 g of dyed cotton fiber was treated with 0.03 g of fixing agent and 30 mL water at 45 °C for 30 min. Then, the dyed cotton fibers were pre-dried at 80 °C for 5 min and cured at 120 °C for 3 min.

### 2.4. Softening Finishing Method

After fixing, the dyed cotton fiber was treated with 0.04 g of softener in 30 mL at room temperature for 15 min. Then the dyed cotton fiber was dried and kept for 12 h at room temperature before measurement.

### 2.5. Color Measurement 

All color measurements were performed using a Datacolor SF600X spectrophotomete (D65 illuminant, specular included, 10° observer angle). The color data was the average of five different places on the sample, and the color yield (*K*/*S*) value was calculated by using the Kubelka-Munk Equation.
*K/S* = (1 − *R*^2^)/2*R*(1)
where *R* is the reflectance of the dyed cotton fiber sample at the λ*_max_* absorption and *K* and *S* are the absorption and scattering coefficients, respectively.

### 2.6. Rubbing Fastness

Rubbing fastness was evaluated by a white fabric (100 mm × 40 mm) against the dyed cotton fiber under wet and dry conditions, using Crocmeter 238 A of SDLA (Shirley Development Laboratories Atlas Inc., Boston, MA, USA) [17]. After rubbing, the *b** value of stained white fabric was measured with Datacolor SF600X spectrophotomete (Datacolor, Lawrenceville, NJ, USA) at the maximum absorption wavelength. The rubbing fastness of sample was evaluated by the *b** value of stained white fabric, because the higher of the *b** value, the bluer the color of the stained white fabric, and the worse the rubbing fastness of dyed cotton fiber.

### 2.7. Friction Coefficient Analysis of Cotton Fiber

Friction coefficient of cotton fiber was investigated with the XCF-1A fiber friction tester (1 CN of tension clamp load, 30 rpm of friction roller speed and 10 mm/min of friction roller falling speed, Shanghai, China) in 10 s. The final test result was the average of 30 fibers [18].

### 2.8. Scanning Electron Microscope Analysis (SEM)

SEM analysis were performed using a VLTRA55 (Carl Zeiss SMT Pte Ltd., Oberkochen, Germany). Cotton fibers were directly flat mounted on aluminum SEM stubs, and adhered by two-sided adhesive film, then covered by a thin gold coating. The microscope accelerating voltage was 1.50 kV, and the beam spot size was 15.

## 3. Results and Discussion

### 3.1. Dyeing Performance of Indigo in Silicon Non-Aqueous Dyeing System

In a traditional water base, indigo was padded with different cycles (1,2,3…12). As shown in Figure 2, the *K*/*S* value of dyed cotton fiber increased with the dyeing cycles, and the *b** value of white fabric was decreased. The smaller the *b** value, the deeper color depth of white fabric. Therefore, more of indigo would be rubbed from the dyed cotton fiber, indicating that the rubbing fastness of dyed cotton fiber was poor. From the results of Figure 2, it can be seen that the *K*/*S* value of dyed cotton fiber was 20.98 and 20.10 when the dyeing cycle was 10 and 12, respectively. Therefore, the color depth of dyed cotton fiber no longer increased when the dyeing cycle was increased to 10. When the dyeing cycle was smaller, indigo-reduced solution would diffuse into the inner of cotton fiber [19], resulting that the surface of dye was lower, so the *b** value of white was higher, and the rubbing fastness of dyed cotton fiber was well. Indigo-reduced solution no longer penetrated into the inner of fiber as the diffusion reached equilibrium. Therefore, more of indigo would adsorb on the surface of the fiber, and indigo which had van der Waals force and hydrogen bonds to the fiber surface was easily washed away [20].

From Table 1, it can be concluded that the *K*/*S* value of dyed cotton fiber was 20.10 when the dyeing cycle was 12 in traditional water base. However, the *K*/*S* value was 25.94 after one dyeing in the silicon non-aqueous dyeing system. Moreover, the *L** and *b** values of cotton fibers which were dyed with the indigo in a traditional water bath were higher than that in silicon non-aqueous medium dye system. Therefore, it is easily to obtain the ideal dyeing depth after one dyeing in the silicon non-aqueous medium dyeing system due to the very strong affinity of indigo-reduced solution to cotton fiber. However, according to the AATCC Test Method 8-2007 [21], the rubbing fastness was worse than in a traditional water base (leve 2~3 in silicon non-aqueous dyeing system vs. level 3 in water base). The reason maybe that the one dyeing would influence the roughness of dyed cotton fiber which will affect the rubbing fastness of dyed cotton fiber. 

### 3.2. Influence of Temperature on the Indigo Dyeing Performance 

Cotton fibers were dyed with indigo-reduced solution at different temperatures (40, 50, 60, 70, 80 and 90 °C); when the liquid rate was 100% (owf), dyeing time was 60 min, the specific gravity of cotton fiber and indigo-reduced solution was 1:1. As shown in Figure 3a, the *K*/*S* value of dyed cotton fiber increased with the increase of dyeing temperature. The color depth of dyed cotton fiber could reach to the maximum (*K*/*S* value was 25.30) when the dyeing temperature was 70 °C. For the rubbing fastness of dyed cotton fiber (Figure 3b), the *b** value of white fabric was lower when the dyeing temperature was lower. However, the changing of *b** value showed a trend of first increasing and then decreasing with the increase of dyeing temperature. For example, the *b** value of white fabric was −10.78, −5.32 and −9.12 under the dyeing temperature at 40, 70 and 90 °C, respectively. This implies that the rubbing fastness of dyed cotton fiber would decrease under a higher temperature (70 °C vs. 90 °C).

The motion of indigo-reduced solution and the swelling of fiber were not violent at a low temperature, resulting in that most of the indigo-reduced solution was difficult to penetrate into the inner of fiber [22,23,24]. Moreover, the reduced solution was oxidized on the fiber surface, and indigo was easily washed off. Therefore, the *K*/*S* value of the dyed cotton fiber was smaller, and the rubbing fastness of dyed cotton fiber was poor when the dyeing temperature was lower. With the increase of dyeing temperature, the swelling of fiber was relatively sufficient, and reduced solution easily penetrated into the inner parts of fiber. However, if the dyeing temperature was too high, not only the dyed cotton fiber would have reddish light, but also a large amount of loose color of dye would occurr on the fiber surface [25]. Therefore, a better dyeing performance of indigo could be achieved in a silicon non-aqueous media dyeing system when the dyeing temperature was 70 °C.

### 3.3. Influence of Dyeing Cycles on the Indigo Dyeing Performance in Silicon Non-Aqueous Medium Dyeing System 

Traditional indigo dyeing needs a multiple pad dyeing (6~10 cycles) to achieve the desired color depth. However, cotton fiber which was dyed with indigo can get a deeper color depth after one dyeing in silicon non-aqueous medium dyeing system. In order to investigate whether indigo could be dyed for multiple cycles in a silicon non-aqueous medium dyeing system, the cotton fibers were dyed for 1~3 cycles. As shown in Figure 4a, the color depth of the dyed cotton fiber was not significantly improved as the increase of dyeing cycle. However, the *b** value of stained white fabric decreased obviously. That is, the rubbing fastness of dyed cotton fiber decreased as the increase of dyeing cycle in the silicon non-aqueous dyeing system.

In the silicon non-aqueous dyeing system, since the indigo-reduced solution was totally incompatible with silicon non-aqueous media, there was a strong affinity between the cotton fiber and the reduced solution because cotton fiber was the hydrophilic fiber. Therefore, nearly 100% of reduced solution can diffuse to the surface of the cotton fiber under the pump force [15]. After one dyeing, some reduced solution was added for the next dyeing. Firstly, the reduced solution must penetrate the indigo which had adsorbed on the surface of the fiber, then it could diffuse to the inner part of the fiber. Compared with traditional water dyeing, there is no padding in a silicon non-aqueous dyeing system, which may influence the arrangement of indigo particles on the fiber surface [26]. Therefore, the rubbing fastness of dyed cotton fiber would decrease with the increase of dyeing cycle. 

### 3.4. Effect of Fixing Finishing on the Rubbing Fastness of Dyed Cotton Fiber

The effect of fixing finishing on the rubbing fastness of dyed cotton fiber is shown in Figure 5. The results showed that fixing finishing had little effect on the *K*/*S* value, but it significantly improved the rubbing fastness of dyed cotton fiber. Obviously, all fixing agents can improve the rubbing fastness of dyed cotton fiber. Compared with unfixed dyed cotton fiber, fixing agent B greatly improve the rubbing fastness of dyed cotton fiber because the *b** value of white fabric was increased to −0.98. For the fixing agent C, it had a little effect on the rubbing fastness of dyed cotton fiber because the *b** value of white fabric was −4.37 after treatment [27].

Comparing the coefficient of friction of dyed cotton fiber and control sample (Table 2), it was found that the dynamic (*u*_a_) and static coefficient (*u*_s_) of friction was increased after dyeing. For example, the dynamic coefficient of friction was increased from 0.0163 to 0.0197, indicating that the fiber surface became rougher after dyeing. The reason maybe that the high concentration of indigo reducing solution was easy agglomerated and oxidized into indigo particles on the surface of fiber [27]. Compared with the dyed cotton fiber, the dynamic and static coefficient of friction of fiber was decreased after fixing, indicating that fiber surface became smoother after fixing.

In order to more intuitively study the reasons for the change of coefficient friction of fiber after fixing, the optical morphology of cotton fiber was observed by using a polarizing microscope [28,29]. As shown in Figure 6, there were some tiny particles on the surface of the dyed cotton fiber. After treating with fixing agents, most of these tiny particles disappeared. Especially, the particles on the cotton fiber surface almost completely disappeared when it was treated with fixing agent B (Figure 6c). According to the observation of single fiber by scanning electron microscopy, the surface of dyed cotton fiber was much rougher, and a large number of uniform particles were loaded on the fiber surface. According to the dyeing environment in the silicon non-aqueous dyeing system, a high concentration of indigo-reduced solution diffuses to the fiber surface, and some molecules of indigo-reduced solution may form aggregates during oxidization, which make the fiber surface rougher. After fixing, some aggregates of indigo particles were washed off, and indigo particles could be rearranged. Therefore, indigo dyes were difficult to rub down after fixing. Moreover, fixing agent B (Figure 6g) had an excellent film-forming property on the fiber surface than the other fixing agents, thus fixing agent B might have a better influence on the rubbing fastness of dyed cotton fiber. The polyurethane-based fixing agents can self-crosslink on the surface of the fiber to form a membrane [30,31,32], which coated the dye and reduced the friction coefficient of fiber. As a result, the polyurethane fixing agents can significantly improve the rubbing fastness of the dyed cotton fiber in an indigo/silicon non-aqueous dyeing system.

### 3.5. Effect of Soften Finishing on the Rubbing Fastness of Dyed Cotton Fiber

After fixing, the dyed cotton fibers were treated with some softeners. The *K*/*S* value, rubbing fastness and handle of the cotton fibers are shown in Figure 7 and Table 3.

As can be seen, fixing finishing and soften finishing had little effect on the *K*/*S* value of dyed cotton fiber [33,34]. A small amount of fixing agents can improve the handle of cotton fiber after treating with softener 899. Comparing Figure 7 with Figure 5, it can be seen that the rubbing fastness of the softened cotton fiber was lower than the fixed cotton fiber, indicating that the soft finishing would influence the fixing effect. The reason may be that the polyurethane membrane would be damaged during soften finishing [33]. This specific mechanism of action of softener on fixing agent will be systemically studied to guide the selection of suitable softeners in industrial production.

## 4. Conclusions

In this paper, we have systemically studied the indigo dyeing parameters in the silicon non-aqueous dyeing system. Compared with traditional water dyeing, although the rubbing fastness level of dyed cotton fiber was 2~3, one deep dyeing could be achieved in the silicon non-aqueous dyeing system (one vs. twelve). In order to improve the rubbing fastness of dyed cotton fiber, the effect of temperature and number of dyeing cycles was studied. The dyeing performance results showed that the optimum temperature for obtaining good rubbing fastness of cotton fiber was about 70 °C, and the optimal dyeing cycle was 1. Meanwhile, the fixing agent can improve the rubbing fastness of dyed cotton fiber significantly, and the best fixing effect on cotton fiber was obtained by cationic waterborne polyurethane. Although soft finishing would weaken the effect of fixing finishing effect on the dyed cotton fiber, the softener can significantly improve the handle of the cotton fiber, because the spinning manufacturing has a certain requirement for the softness of dyed cotton fiber.

## Figures and Tables

**Figure 1 polymers-11-01854-f001:**
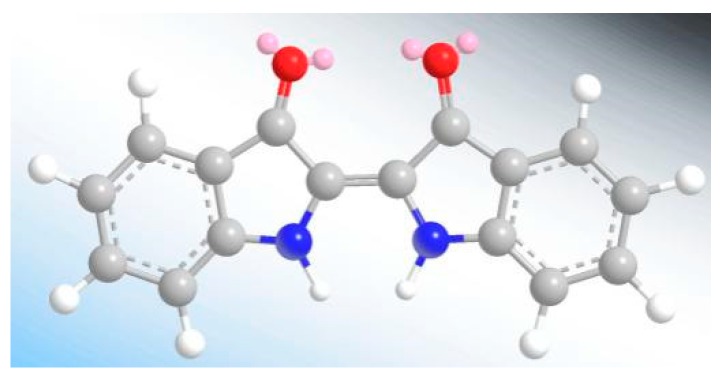
3D molecular structure of indigo.

**Figure 2 polymers-11-01854-f002:**
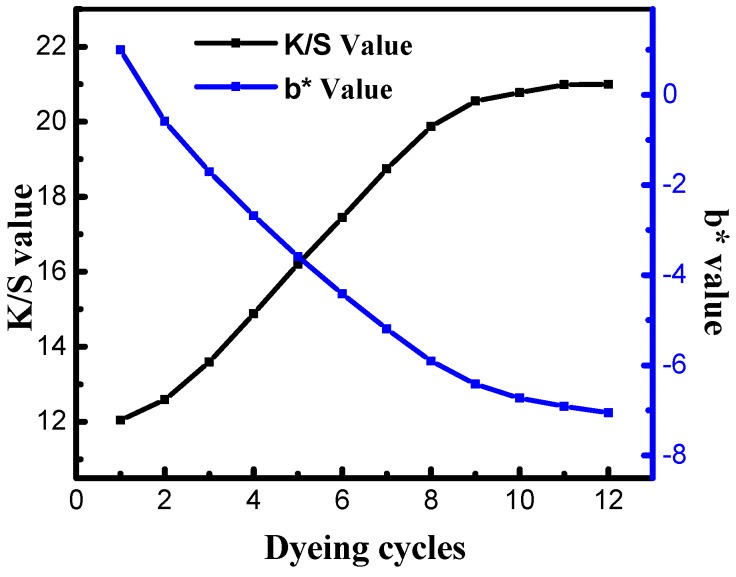
*K*/*S* value of dyed cotton fiber and *b** value of stained white fabric for indigo dyeing in traditional water base.

**Figure 3 polymers-11-01854-f003:**
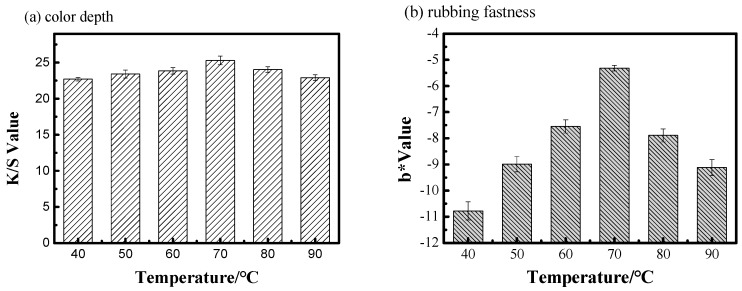
Effect of temperature on the color depth (**a**) and rubbing fastness of dyed cotton fiber (**b**).

**Figure 4 polymers-11-01854-f004:**
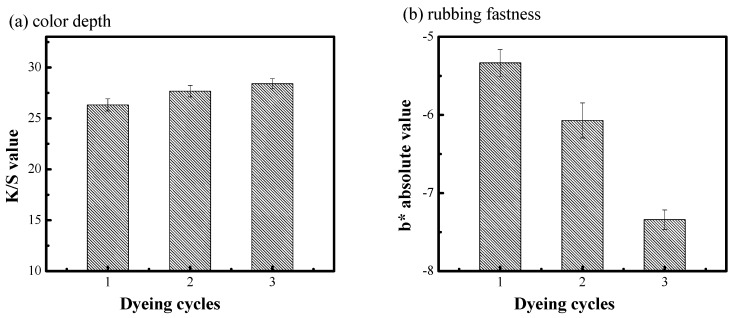
Effect of dyeing cycles on the color depth (**a**) and rubbing fastness of dyed cotton fiber (**b**).

**Figure 5 polymers-11-01854-f005:**
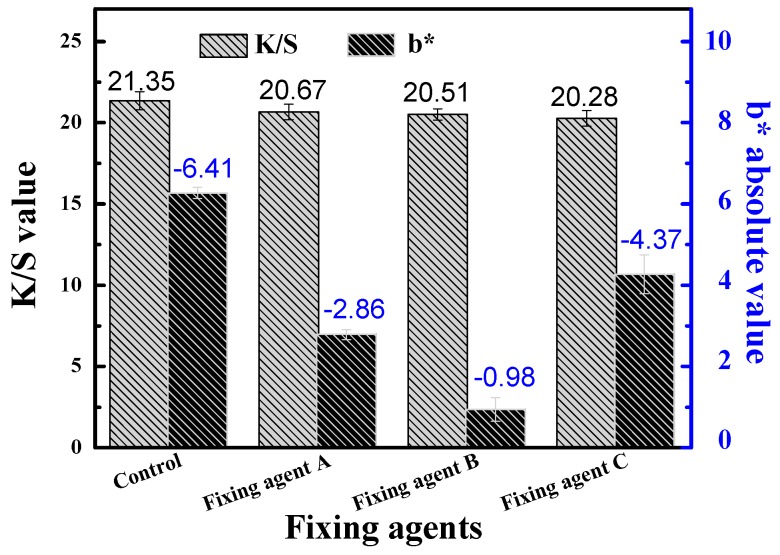
*K*/*S* value and rubbing fastness of fixed cotton fiber.

**Figure 6 polymers-11-01854-f006:**
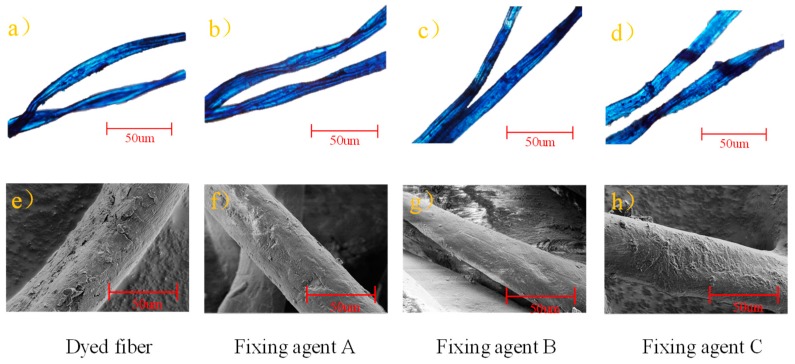
The optical morphology and SEM image of cotton fiber after fixing: (**a**) and (**e**) were the dyed control samples;(**b**) and (**f**) were the dyed fibers treating with fixing agent A; (**c**) and (**g**) were the dyed fibers treating with fixing agent B; (**d**) and (**h**) were the dyed fibers treating with fixing agent C.

**Figure 7 polymers-11-01854-f007:**
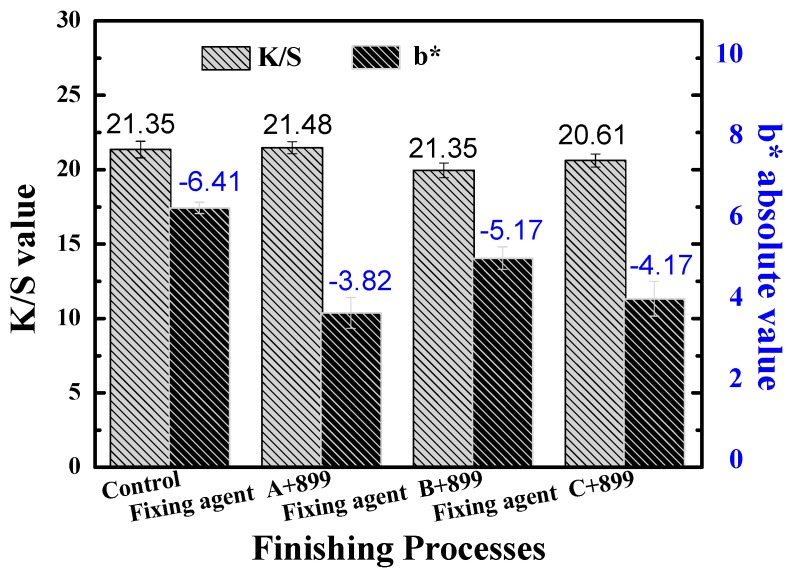
The *K*/*S* value and rubbing fastness of softened cotton fiber.

**Table 1 polymers-11-01854-t001:** The dyeing performance of indigo in silicon dyeing system and traditional water base.

Dyeing System	Dyeing Cycle	*K*/*S* Value	Rubbing Fatness	*L**	*a**	*b** (Cotton Fiber)
water	12	20.10	3	20.79	2.84	−10.64
silicon	1	25.94	2~3	17.39	2.34	−14.92

**Table 2 polymers-11-01854-t002:** The coefficient of friction on cotton fiber after fixing finishing.

	Control Sample	Dyed Cotton Fiber	Fixing Agent A	Fixing Agent B	Fixing Agent C
*f* _s_	57.20	65.51	63.74	61.31	63.86
*u* _s_	0.0196	0.0224	0.0214	0.0212	0.0216
*f* _a_	52.72	60.57	57.59	56.31	59.03
*u* _a_	0.0163	0.0197	0.0191	0.0189	0.0194

*f*_s_ means that static friction; *u*_s_ means that static coefficient; *f*_a_ means that dynamic friction; *u*_a_ means that dynamic coefficient.

**Table 3 polymers-11-01854-t003:** The handle of the cotton fiber after fixing finishing and soften finishing.

Sample	Dyed Cotton Fiber	Fixing Finishing			Fixing Finishing and Soften Finishing (899)		
		Fixing Agent A	Fixing Agent B	Fixing Agent C	Fixing Agent A	Fixing Agent B	Fixing Agent C
**Handle**	rough	rough	soft	rough	soft	soft	soft

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
