# Peer review of "Improvement of the Rubbing Fastness of Cotton Fiber in Indigo/Silicon Non-Aqueous Dyeing Systems"

_polymers, 2019, doi:10.3390/polym11111854_

Round 1

Reviewer 1 Report

I recommend to accept after minor revision.

This paper is very compact script and focus on the non aqueous dyeing system. It's good theme about ecofriendly dyeing systems.

Just one point out,
In the table 2, there are used the abbreviation like fs, us, fa, and ua. Where is define? If not, add to the manuscripts.

Author Response

Point 1: In the table 2, there are used the abbreviation like fs, us, fa, and ua. Where is define? If not, add to the manuscripts.

Response 1: We appreciate the reviewer’s comment. We have followed the suggestion and defined some abbreviations in manuscript.

Change: ...Compared the coefficient of friction of dyed cotton fiber and control sample (Table 2), it was found that the dynamic (ua) and static coefficient (us) of friction was increased after dyeing.

Table 2. The coefficient of friction on cotton fiber after fixing finishing.

Control sample

Dyed cotton fiber

Fixing agent A

Fixing agent B

Fixing agent C

fs

57.20

65.51

63.74

61.31

63.86

us

0.0196

0.0224

0.0214

0.0212

0.0216

fa

52.72

60.57

57.59

56.31

59.03

ua

0.0163

0.0197

0.0191

0.0189

0.0194

fs means that static friction; us means that static coefficient; fa means that dynamic friction; ua means that dynamic coefficient 

...and the color yield (K/S) value was calculated by using the Kubelka-Munk equation.

                       K/S=(1-R2)/2R                           (1)

where R is the reflectance of the dyed cotton fiber sample at theλmax absorption and K and S are the absorption and scattering coefficients, respectively.

Reviewer 2 Report

The authors have used different dying medium to increase the rubbing fastness and depth of coloring in cotton fabric. This article presents a more practical application of the knowledge on useful daily needs. Therefore the article is suited for the Journal, polymers 

The terms such as K/S should be defined at the first use. Other quantities are b* - This is necessary for publishing the manuscript. The term dying times is confusing. It is highly recommended to change the term dying times to 'dying cycles' or any other term that that fits the context. In the discussion, the authors have given mutually contradicting phrases about the dying depth and the b* value. In the finishing section, the relevant literature where some non conventional fixing agents are applied to pure cellulose (paper) can also be cited. Biosensors and Bioelectronics 126, 831-837 Can the author give a valuable reason for the best rubbing fastness at 70 C ? 

Author Response

Point 1: The terms such as K/S should be defined at the first use. Other quantities are b* - This is necessary for publishing the manuscript.

Response 1: We appreciate the reviewer’s comment. We have followed the suggestion and defined some abbreviations in manuscript. The b* value is one of the coordinates of psychometric chroma coordinates, and the other coordinates are a* and L* value. From the following Figure, it can be concluded that the smaller the b* value, the bluer the color of sample.

Change: ...and the color yield (K/S) value was calculated by using the Kubelka-Munk equation.

                       K/S=(1-R2)/2R                           (1)

where R is the reflectance of the dyed cotton fiber sample at theλmax absorption and K and S are the absorption and scattering coefficients, respectively.

The rubbing fastness of sample was evaluated by the b* value of stained white fabric, because the higher of the b* value, the bluer the color of the stained white fabric, and the worse the rubbing fastness of dyed cotton fiber.

Point 2: The term dying times is confusing. It is highly recommended to change the term dying times to 'dying cycles' or any other term that that fits the context.

Response2: We appreciate the reviewer’s comment. We have followed the suggestion and modified the ‘dyeing times’ to ‘dyeing cycles’. 

Change: ...fiber increased with the padding cycles...

it showed that the K/S value of dyed cotton fiber was balanced when the dyeing cycle was 10. When the dyeing cycle was smaller...

Traditional indigo dyeing needs a multiple pad dyeing (6~10 cycles) to achieve...

Table 1. Compared with the rubbing fatness of dyed cotton fiber in silicon dyeing system and traditional water base.

Dyeing system

Dyeing cycle

K/S value

Rubbing fatness

L*

a*

b*

water

12

20.10

3

20.79

2.84

-10.64

silicon

1

25.94

2~3

17.39

2.34

-14.92

Figure 2. K/S value of dyed cotton fiber and b* value of stained white fabric for indigo dyeing in traditional water base.

Figure 4. Effect of dyeing cycles on the color depth (a) and rubbing fastness of dyed cotton fiber (b).

Point 3: In the discussion, the authors have given mutually contradicting phrases about the dying depth and the b* value.

Response 3: We appreciate the reviewer’s comment. After dyeing, K/S value of dyed cotton fiber was employed to represent the color depth. However, the b* value was not the value of dyed cotton fiber, its belonged to the stained white fabric, which was used to rub the dyed cotton fiber. Only the b* value was represented the dyed cotton fiber in Table 1.

Change: ...The rubbing fastness of sample was evaluated by the b* value of stained white fabric, because the higher of the b* value, the bluer the color of the stained white fabric, and the worse the rubbing fastness of dyed cotton fiber.

Table 1. The dyeing performance of indigo in silicon dyeing system and traditional water base.

Dyeing system

Dyeing cycle

K/S value

Rubbing fatness

L*

a*

b*(cotton fiber)

water

12

20.10

3

20.79

2.84

-10.64

silicon

1

25.94

2~3

17.39

2.34

-14.92

Point 4: In the finishing section, the relevant literature where some non conventional fixing agents are applied to pure cellulose (paper) can also be cited. Biosensors and Bioelectronics 126, 831-837

Response 4: We appreciate the reviewer’s comment. We have followed the suggestion and add some references about fixing agents which are applied to pure cellulose.

Change: 

Correia,M. S., Miranda, A. S., Oliveira, M. C., et al. Analysis of friction in the ejection of thermoplastic mouldings. Int. J. Adv. Manuf. Tech2012, 59. Jeevarathinam, A. S.;Pai, N.; Huang, K.; Hariri, A.; Wang, J.; Bai, Y.; Jokerst, J. V.; et al. A cellulose-based photoacoustic sensor to measure heparin concentration and activity in human blood samples. Biosens. Bioelectron. 2019, 126, 831-837.

...

Point 5: Can the author give a valuable reason for the best rubbing fastness at 70 oC ?  

Response 5: We appreciate the reviewer’s comment. We have followed the suggestion and add some discussion in manuscript.

Change: ...However, not only the dyed cotton fiber would have reddish light, but also a large amount of loose color of dye would be occurred on the fiber surface [24]. Therefore, a better dyeing performance of indigo could be achieved in silicon non-aqueous media dyeing system when the dyeing temperature was 70 ℃.

Reviewer 3 Report

The paper shows the effect of temperature, dyeing frequency, fixing finishing and soften finishing on  rubbing fastness  after the application of a new indigo dyeing technology. The paper shows some interesting results however I don’t feel to suggest publication unless English language is revised and the following points are clarified or better addressed:

I suggest to change “cowboy” into “cowboy wear” on line 24 It would be useful to the reader that the nature of the siloxane non-aqueous media, fixing agents A, B and C and softener would be better addressed in terms of composition  (lines 62, 65 and 67) If the “ multiple deying process” described on lines 80-84 is used in the part of the work described in section 3.3 this should be reminded.  If I correctly interpreted the test  the iterative process would be repeated with concentrations of the deyng solution 1/3 the one used in the single step procedure and left to  partially oxidize.  Please clarify and, if the interpretation is correct, explain the reasons  of   this choice Ref. 17 (line 105) cited in the experimental section relative to rubbing fastness doesn’t deal with this technique. However it would be useful for the reader a few words about the meaning of the rubbing fastness values reported in table 1: how is determined a rubbing value 2 or 3? On lines 106-108 the authors say: “The rubbing fastness of sample was evaluated by the b* value of stained white fabric, because the higher of the b* value, the worse the rubbing fastness of dyed cotton fiber”. Shouldn’t be the contrary (so as reported on lines 122-124)? Ref.18 (line 112) has any bearing with friction coefficient analysis. Please, however, clarify the meaning of fs, us, fa, ua in table 2. I understand that the “different times” on line 120 are “different number of deying- washing cycles” described on lines  85-88. If this is right  please, for the better understanding, speak of number of cycles or similar expressions. On lines 124-125 the authors say: “…the K/S value of dyed cotton fiber was balanced when the dyeing time was 10”. Do the authors mean that a plateau value is reached? On line 126 it is said: “….resulting that the surface of dye was low….”. Do the authors mean “ the dye on the surface....”? Table 1 shows more data than indicated in the legend. Please detail better. Does dyeing time mean number of deying cycles? Please clarify in the legend. On lines 151-152 the authors say : “This implies that the rubbing fastness of dyed cotton fiber would decrease under a higher temperature.” Does it refer to the change from 70 to 90°C? In this case the b* value increase shouldn’t imply a progressively better rubbing fastness? On lines 153-159 the authors give an explanation without taking into account that the histogram of fig.3b depicts a non monotonous trend with a minimum. Please clarify On line 162 the title “influence of deying times…” should be changed into “the influence of number of deying- washing cycles…..”. The same on lines 167and 169 and 178 Please clarify the statement “Whether a multiple dyeing can be achieved in silicon non-aqueous dyeing system.”(lines 165-166) The reviewer understands that in Fig.4b the values are all negative. If this is true the histogram is quite unusual. I suggest to represent the absolute values and indicate in the legend that they are all negative. However if they are all negative the b* value would decrease with the number of deyng cycles contrarily to the statement of the authors: However, the b* value of white fabric increased obviously (lines 167-168) . Figg. 5 and 7 show hystograms with negative values  so as Fig.4b. Please modify so as suggested at previous point What makes the authors sure that (lines 172-173) : “….nearly 100% of reduced solution can diffuse to the surface of cotton fiber under mechanical force “ ? Couldn’t an emulsion in the siloxane non-aqueous media be formed? Which are the “mechanical forces” they refer to? On lines 236-237 the statement “dyeing temperature and dyeing time were studied” should be adjusted into “the effect of temperature and number of dyeing- washing cycles was studied” About the conclusion “optimum temperature for obtaining good rubbing fastness of cotton fiber was about 70 °C and” (line 238) see the observation at point 11th

Author Response

Response to Reviewer 3 Comments

Point 1: I suggest to change “cowboy” into “cowboy wear” on line 24.

Response 1: We appreciate the reviewer’s comment. We have followed the suggestion and changed ‘cowboy’.

Change: Most of the traditional cowboy wear are made of warp yarns...

Point 2: It would be useful to the reader that the nature of the siloxane non-aqueous media, fixing agents A, B and C and softener would be better addressed in terms of composition  (lines 62, 65 and 67).

Response 2: We appreciate the reviewer’s comment. We have followed the suggestion and added the compositions of these materials.

Change: Siloxane non-aqueous media (decamethyl cyclopentasiloxane, purity>98%) was purchased from GE Toshiba Silicone Ltd... Fixing agent A (acrylate polymer), B (cationic waterborne polyurethane) and C (non-ionic waterborne polyurethane) were industrial grade...

Point 3: If the “ multiple deying process” described on lines 80-84 is used in the part of the work described in section 3.3 this should be reminded. If I correctly interpreted the test the iterative process would be repeated with concentrations of the deyng solution 1/3 the one used in the single step procedure and left to  partially oxidize.  Please clarify and, if the interpretation is correct, explain the reasons  of   this choice Ref. 17 (line 105) cited in the experimental section relative to rubbing fastness doesn’t deal with this technique. However it would be useful for the reader a few words about the meaning of the rubbing fastness values reported in table 1: how is determined a rubbing value 2 or 3?

Response 3: We appreciate the reviewer’s comment. We have followed the suggestion and clarified the multiple deying process. In silicone non-aqueous media dyeing system, cotton fiber was dyed with indigo, not yarn or fabric. According to Colorfastness to Crocking: AATCC Crockmeter Method, all fibers in the form of yarn or fabric whether dyed, printed or otherwise colored, can be tested. However, the rubbing fastness of dyed cotton fiber can not be tested. Therefore, we used a white fabric to rub the dyed cotton fiber. After rubbing, some of indigo dyes would diffuse to the white fabric. As a result, we used the b* value of stained white fabric to represent the rubbing fastness of dyed cotton fiber, because the higher of the b* value, the bluer the color of the stained white fabric, and the worse the rubbing fastness of dyed cotton fiber. In table 1, according to AATCC Test Method 8-2007, the rubbing fastness value 2 or 3 was obtained.   

Change: Whether a multiple dyeing can be achieved in silicon non-aqueous dyeing system. The multiple indigo dyeing was investigated. The dyeing method was the same as the silicon non-aqueous dyeing system. Firstly, indigo reduced solution was divided into two or three parts. Then, 1/2 or 1/3 of indigo reduced solution was put into dyeing system for 60 min. After oxidization, the remaining 1/2 or 1/3 of the reduced solution was added into dyeing system for another 60 min.

The rubbing fastness of sample was evaluated by the b* value of stained white fabric, because the higher of the b* value, the bluer the color of the stained white fabric, and the worse the rubbing fastness of dyed cotton fiber.

Campbell, B.;Inkumsah, S. E.; Tandoh, W. C. The fastness of indigo and topped indigo dyeings on wool cloth.  Technol. 2010, 80, 583-587. Colorfastness to Crocking: AATCC Crockmeter Method. AATCC Test Method 8-2007.

Point 4: On lines 106-108 the authors say: “The rubbing fastness of sample was evaluated by the b* value of stained white fabric, because the higher of the b* value, the worse the rubbing fastness of dyed cotton fiber”. Shouldn’t be the contrary (so as reported on lines 122-124)?

Response 4: We appreciate the reviewer’s comment. After dyeing, K/S value of dyed cotton fiber was employed to represent the color depth of fiber. However, the b* value was not the value of dyed cotton fiber, its belonged to the stained white fabric, which was used to rub the dyed cotton fiber. Only the b* value was represented the dyed cotton fiber in Table 1.

Change: ...The rubbing fastness of sample was evaluated by the b* value of stained white fabric, because the higher of the b* value, the bluer the color of the stained white fabric, and the worse the rubbing fastness of dyed cotton fiber.

Table 1. The dyeing performance of indigo in silicon dyeing system and traditional water base.

Dyeing system

Dyeing cycle

K/S value

Rubbing fatness

L*

a*

b*(cotton fiber)

water

12

20.10

3

20.79

2.84

-10.64

silicon

1

25.94

2~3

17.39

2.34

-14.92

Point 5: Ref.18 (line 112) has any bearing with friction coefficient analysis. Please, however, clarify the meaning of fs, us, fa, ua in table 2.

Response 5: We appreciate the reviewer’s comment. We have followed the suggestion and defined some abbreviations in manuscript.

Change: ...Compared the coefficient of friction of dyed cotton fiber and control sample (Table 2), it was found that the dynamic (ua) and static coefficient (us) of friction was increased after dyeing.

Table 2. The coefficient of friction on cotton fiber after fixing finishing.

Control sample

Dyed cotton fiber

Fixing agent A

Fixing agent B

Fixing agent C

fs

57.20

65.51

63.74

61.31

63.86

us

0.0196

0.0224

0.0214

0.0212

0.0216

fa

52.72

60.57

57.59

56.31

59.03

ua

0.0163

0.0197

0.0191

0.0189

0.0194

fs means that static friction; us means that static coefficient; fa means that dynamic friction; ua means that dynamic coefficient 

Point 6: I understand that the “different times” on line 120 are “different number of deying- washing cycles” described on lines  85-88. If this is right please, for the better understanding, speak of number of cycles or similar expressions.

Response 6: We appreciate the reviewer’s comment. We have followed the suggestion and modified the ‘dyeing times’ to ‘dyeing cycles’. 

Change: ...fiber increased with the padding cycles...

it showed that the K/S value of dyed cotton fiber was balanced when the dyeing cycle was 10. When the dyeing cycle was smaller...

Traditional indigo dyeing needs a multiple pad dyeing (6~10 cycles) to achieve...

Table 1. Compared with the rubbing fatness of dyed cotton fiber in silicon dyeing system and traditional water base.

Dyeing system

Dyeing cycle

K/S value

Rubbing fatness

L*

a*

b*

water

12

20.10

3

20.79

2.84

-10.64

silicon

1

25.94

2~3

17.39

2.34

-14.92

Figure 2. K/S value of dyed cotton fiber and b* value of stained white fabric for indigo dyeing in traditional water base.

Figure 4. Effect of dyeing cycles on the color depth (a) and rubbing fastness of dyed cotton fiber (b).

Point 7: On lines 124-125 the authors say: “…the K/S value of dyed cotton fiber was balanced when the dyeing time was 10”. Do the authors mean that a plateau value is reached?

Response 7: We appreciate the reviewer’s comment. From Figure 1, the K/S value of dyed cotton fiber was 20.98 and 20.99 when the dyeing cycle was 10 and 12, respectively, indicating that the K/S value of dyed cotton fiber was no longer increased when the dyeing cycle was increased to 10.  

Change: From the results of Figure 2, it showed that the K/S value of dyed cotton fiber was 20.98 and 20.99 when the dyeing cycle was 10 and 12, respectively, indicating that the color depth of dyed cotton fiber was no longer increased when the dyeing cycle was increased to 10. 

Point 8: On lines 124-125 the authors say: “…the K/S value of dyed cotton fiber was balanced when the dyeing time was 10”. Do the authors mean that a plateau value is reached?

Response 8: We appreciate the reviewer’s comment. From Figure 1, the K/S value of dyed cotton fiber was 20.98 and 20.10 when the dyeing cycle was 10 and 12, respectively, indicating that the K/S value of dyed cotton fiber was no longer increased when the dyeing cycle was increased to 10.  

Change: From the results of Figure 2, it showed that the K/S value of dyed cotton fiber was 20.98 and 20.10 when the dyeing cycle was 10 and 12, respectively. Therefore, the color depth of dyed cotton fiber was no longer increased when the dyeing cycle was increased to 10. 

Point 9: Table 1 shows more data than indicated in the legend. Please detail better. Does dyeing time mean number of deying cycles? Please clarify in the legend.

Response 9: We appreciate the reviewer’s comment. Dyeing time mean number of dyeing cycles. We had modified it and analyzed the date in Table 1 more carefully.  

Change: From table 1, it can be concluded that the K/S value of dyed cotton fiber was 20.10 when the dyeing cycle was 12 in traditional water base. However, the K/S value was 25.94 after one dyeing in the silicon non-aqueous dyeing system. Moreover, the L* and b* values of cotton fibers which were dyed with the indigo in traditional water bath were higher than that in silicon non-aqueous medium dye system. Therefore, it is easily to obtain the ideal dyeing depth after one dyeing in the silicon non-aqueous medium dyeing system due to the very strong affinity of indigo reduced solution to cotton fiber. However, according to the AATCC Test Method 8-2007 [21], the rubbing fastness was worse than in traditional water base (lever 2~3 in silicon non-aqueous dyeing system vs. level 3 in water base). The reason maybe that the one dyeing would influence the roughness of dyed cotton fiber which will affect the rubbing fastness of dyed cotton fiber.

Table 1. The dyeing performance of indigo in silicon dyeing system and traditional water base.

Dyeing system

Dyeing cycle

K/S value

Rubbing fatness

L*

a*

b*(cotton fiber)

water

12

20.10

3

20.79

2.84

-10.64

silicon

1

25.94

2~3

17.39

2.34

-14.92

Point 10: On lines 151-152 the authors say : “This implies that the rubbing fastness of dyed cotton fiber would decrease under a higher temperature.” Does it refer to the change from 70 to 90°C? In this case the b* value increase shouldn’t imply a progressively better rubbing fastness?

Response 10: We appreciate the reviewer’s comment. We have followed the suggestion and add some discussion about the dyeing temperature.

Change: ...However, if the dyeing temperature was too high, not only the dyed cotton fiber would have reddish light, but also a large amount of loose color of dye would be occurred on the fiber surface [25]. Therefore, a better dyeing performance of indigo could be achieved in silicon non-aqueous media dyeing system when the dyeing temperature was 70 ℃.

Point 11: On lines 153-159 the authors give an explanation without taking into account that the histogram of fig.3b depicts a non monotonous trend with a minimum. Please clarify.

Response 11: We appreciate the reviewer’s comment. We have followed the suggestion and add some discussion about the dyeing temperature.

Change: ...For the rubbing fastness of dyed cotton fiber (Figure 3b), the b* value of white fabric was higher when the dyeing temperature was lower. However, the changing of he b* value showed a trend of first increasing and then decreasing with the increase of dyeing temperature. For example, the b* value of white fabric was -10.78, -5.32 and -9.12 under the dyeing temperature at 40 oC, 70 oC and 90o C, respectively. This implies that the rubbing fastness of dyed cotton fiber would decrease under a higher temperature (70 oC vs. 90o C).  

Point 12: On line 162 the title “influence of dyeing times…” should be changed into “the influence of number of dyeing-washing cycles…..”. The same on lines 167and 169 and 178 Please clarify the statement “Whether a multiple dyeing can be achieved in silicon non-aqueous dyeing system.”(lines 165-166).

Response 12: We appreciate the reviewer’s comment. We have followed the suggestion and modified the ‘dyeing times’ to ‘dyeing cycles’. 

Change: ...3.3. Influence of dyeing cycles on the indigo dyeing performance  in silicon non-aqueous medium dyeing system 

Traditional indigo dyeing needs a multiple pad dyeing (6~10 cycles) to achieve...

...dyed cotton fiber was not significantly improved as the increase of dyeing cycle... 

...as the increase of dyeing cycle in the silicon non-aqueous dyeing system...

In order to investigate whether indigo could be dyed multiple cycles in silicon non-aqueous medium dyeing system, the cotton fibers were dyed 1~3 cycles. As shown in Figure 4(a),...

Point 13: (lines 165-166) The reviewer understands that in Fig.4b the values are all negative. If this is true the histogram is quite unusual. I suggest to represent the absolute values and indicate in the legend that they are all negative. However if they are all negative the b* value would decrease with the number of deyng cycles contrarily to the statement of the authors: However, the b* value of white fabric increased obviously (lines 167-168) . Fig. 5 and 7 show hystograms with negative values  so as Fig.4b. Please modify so as suggested at previous point.

Response 13: We appreciate the reviewer’s comment. We used the b* absolute values of stained white fabric to indicate the rubbing fastness of dyed cotton fiber. In Figure 4b, the b* value was not the value of dyed cotton fiber, its belonged to the stained white fabric, which was used to rub the dyed cotton fiber. After rubbing, some of indigo dyes would abscise from dyed cotton fiber, and diffuse to the stained white fabric. Therefore, we used the b* value of white fabric to indicate the rubbing fastness of dyed cotton fiber, because the higher of the b* value, the bluer the color of the stained white fabric, and the worse the rubbing fastness of dyed cotton fiber.

Change: ...The rubbing fastness of sample was evaluated by the b* value of stained white fabric, because the higher of the b* value, the bluer the color of the stained white fabric, and the worse the rubbing fastness of dyed cotton fiber.

Figure 4. Effect of dyeing cycles on the color depth (a) and rubbing fastness of dyed cotton fiber (b).

Figure 5. K/S value and rubbing fastness of fixed cotton fiber.

Figure 7. The K/S value and rubbing fastness of softened cotton fiber.

Point 14: What makes the authors sure that (lines 172-173) : “….nearly 100% of reduced solution can diffuse to the surface of cotton fiber under mechanical force “ ? Couldn’t an emulsion in the siloxane non-aqueous media be formed? Which are the “mechanical forces” they refer to?

Response 14: We appreciate the reviewer’s comment. In silicon non-aqueous medium dyeing system, since the indigo leuco is insoluble in non-aqueous medium but has a high affinity to cotton fiber and the non-aqueous medium has good isolation effect and can prevent the oxidation of the leuco [15], a dark indigo dyeing of cotton fiber with 90% dyeing rate can be obtained by one dyeing. Surfactant was not used during dyeing, so an emulsion was not formed in this dyeing system. The pump must be employed in the dyebath recycling, therefore, the ‘mechanical force’ refer to the pump force.

Change: ...In silicon non-aqueous dyeing system, since indigo reduced solution was totally incompatible with silicon non-aqueous media, but there was a strong affinity between the cotton fiber and the reduced solution because cotton fiber was the hydrophilic fiber. Therefore, nearly 100% of reduced solution can diffuse to the surface of cotton fiber under the pump force [15].

Point 15: On lines 236-237 the statement “dyeing temperature and dyeing time were studied” should be adjusted into “the effect of temperature and number of dyeing- washing cycles was studied”.

Response 15: We appreciate the reviewer’s comment. We have modified the ‘dyeing temperature and dyeing time were studied to ‘the effect of temperature and number of dyeing cycles was studied’.

Change: ...the effect of temperature and number of dyeing cycles was studied...

Point 16: About the conclusion “optimum temperature for obtaining good rubbing fastness of cotton fiber was about 70 °C and” (line 238) see the observation at point 11th.a

Response 16: We appreciate the reviewer’s comment. We have interpreted this sentence.

Change: ...The dyeing performance results showed that the optimum temperature for obtaining good rubbing fastness of cotton fiber was about 70 °C and the optimal dyeing cycle was 1. 

Round 2

Reviewer 2 Report

The authors are adviced to check for English corrections thoroughly. The authors have addressed the comments satisfactorily and the manuscript may be published after English corrections for grammar and spellings. 

Reviewer 3 Report

The observatuions  were all clarified or taken into account. I feal to suggest publication.